# New Alkylpyridinium Anthraquinone, Isocoumarin, C-Glucosyl Resorcinol Derivative and Prenylated Pyranoxanthones from the Culture of a Marine Sponge-Associated Fungus, *Aspergillus stellatus* KUFA 2017

**DOI:** 10.3390/md20110672

**Published:** 2022-10-27

**Authors:** Fátima P. Machado, Inês C. Rodrigues, Luís Gales, José A. Pereira, Paulo M. Costa, Tida Dethoup, Sharad Mistry, Artur M. S. Silva, Vitor Vasconcelos, Anake Kijjoa

**Affiliations:** 1ICBAS-Instituto de Ciências Biomédicas Abel Salazar, Rua de Jorge Viterbo Ferreira, 228, 4050-313 Porto, Portugal; 2Interdisciplinary Centre of Marine and Environmental Research (CIIMAR), Terminal de Cruzeiros do Porto de Leixões, Av. General Norton de Matos s/n, 4450-208 Matosinhos, Portugal; 3Instituto de Biologia Molecular e Celular (i3S-IBMC), Universidade do Porto, Rua de Jorge Viterbo Ferreira, 228, 4050-313 Porto, Portugal; 4Department of Plant Pathology, Faculty of Agriculture, Kasetsart University, Bangkok 10240, Thailand; 5Department of Chemistry, University of Leicester, University Road, Leicester LE 7 RH, UK; 6Departamento de Química & QOPNA, Universidade de Aveiro, 3810-193 Aveiro, Portugal; 7FCUP-Faculty of Sciences, University of Porto, Rua do Campo Alegre, s/n, 4169-007 Porto, Portugal

**Keywords:** *Aspergillus stellatus*, Trichocomaceae, marine sponge-associated fungus, anthraquinones, isocoumarin, C-glucosyl resorcinols, antibacterial activity, antibiofilm activity

## Abstract

An unreported isocoumarin, (*3S*,4*R*)-4-hydroxy-6-methoxymellein (**2**), an undescribed propylpyridinium anthraquinone (**4**), and an unreported C-glucosyl resorcinol derivative, acetyl carnemycin E (**5c**), were isolated, together with eight previously reported metabolites including *p*-hydroxybenzaldehyde (**1**), 1,3-dimethoxy-8-hydroxy-6-methylanthraquinone (**3a**), 1,3-dimethoxy-2,8-dihydroxy-6-methylanthraquinone (**3b**), emodin (**3c**), 5[(3*E*,5*E*)-nona-3,5-dien-1-yl]benzene (**5a**), carnemycin E (**5b**), tajixanthone hydrate (**6a**) and 15-acetyl tajixanthone hydrate (**6b**), from the ethyl acetate extract of the culture of a marine sponge-derived fungus, *Aspergillus stellatus* KUFA 2017. The structures of the undescribed compounds were elucidated by 1D and 2D NMR and high resolution mass spectral analyses. In the case of **2**, the absolute configurations of the stereogenic carbons were determined by comparison of their calculated and experimental electronic circular dichroism (ECD) spectra. The absolute configurations of the stereogenic carbons in **6a** and **6b** were also determined, for the first time, by X-ray crystallographic analysis. Compounds **2**, **3a**, **3b**, **4**, **5a**, **5b**, **5c**, **6a**, and **6b** were assayed for antibacterial activity against four reference strains, viz. two Gram-positive (*Staphylococcus aureus* ATCC 29213, *Enterococcus faecalis* ATCC 29212) and two Gram-negative (*Escherichia coli* ATCC 25922, *Pseudomonas aeruginosa* ATCC 27853), as well as three multidrug-resistant strains. However, only **5a** exhibited significant antibacterial activity against both reference and multidrug-resistant strains. Compound **5a** also showed antibiofilm activity against both reference strains of Gram-positive bacteria.

## 1. Introduction

The genus *Aspergillus* (family Aspergillaceae) is one of the most extensively studied genera of filamentous fungi, mainly due to the medical relevance, food spoilage, and industrial application of some of its species. Aspergilli can grow in a wide range of niches, mainly in soils and on dead matter, but some are also capable of colonizing living animal or plant hosts. However, the increasing numbers of *Aspergillus* species have been found in different niches in the marine environment, from tropical waters [1] to the Arctic [2], and from shallow waters [3] to the deep-sea [4]. In total, approximately 350 *Aspergillus* species have been identified [5].

In the last two decades, marine-derived *Aspergillus* species have attracted tremendous attention from researchers working on marine natural products since they were responsible for a production of the highest numbers of marine fungal metabolites as demonstrated by two review articles, one covering the period of January 1992 to December 2014, reporting 512 metabolites, and another covering the period of 2015 to December 2020, which reported 361 compounds, isolated from marine-derived *Aspergillus* species [6]. It is also worth mentioning that the specialized metabolites produced by marine-derived *Aspergillus* species possess not only structural diversity such as indole alkaloids, diketopiperazine derivatives, meroterpenoids, anthraquinones, isocoumarins, xanthones, *p*-terphenyl derivatives, and peptides, but also a myriad of biological activities [6].

In our pursuit of antibiotic and antibiofilm compounds from marine-derived fungi from tropical seas, we focused our attention on the marine sponge-associated *Aspergillus stellatus* since this fungus has not been extensively investigated. A literature search revealed that a mycotoxin, asteltoxin, was isolated from toxic maize meal cultures of *A. stellatus* Curzi (MRC 277) [7]. In another work, Kamal et al. reported the isolation of tajixanthone, shamixanthone, ajamxanthone, shahenxanthone, najamxanthone, radixanthone, and mannitol from the mycelium of *A. stellatus* Curzi [8]. For this reason, we investigated secondary metabolites from the culture of *A. stellatus* KUFA 2017, isolated from a marine sponge, *Mycale* sp., which was collected from the coral reef at the Samaesan Island, in the Gulf of Thailand.

Fractionation of the ethyl acetate (EtOAc) extract of the culture *A. stellatus* KUFA 2017 by column chromatography of silica gel, followed by purification by preparative TLC, Sephadex LH-20 column and crystallization, led to the isolation of undescribed (*3S*,4*R*)-4-hydroxy-6-methoxymellein (**2**), stellatanthraquinone (**4**), and acetyl carnemycin E (**5c**), as well as the previously reported *p*-hydroxybenzaldehyde (**1**) [9], 1,3-dimethoxy-8-hydroxy-6-methylanthraquinone (**3a**) [10], 1,3-dimethoxy-2,8-dihydroxy-6-methylanthraquinone (**3b**) [11], emodin (**3c**) [12] and 5[(3*E*,5*E*)-nona-3,5-dien-1-yl]benzene (**5a**) [13], carnemycin E (**5b**) [13], tajixanthone hydrate (**6a**) [14,15], and 15-acetyl tajixanthone hydrate (**6b**) [16]. (Figure 1). The structures of the undescribed compounds were established on the basis of an extensive analysis of their 1D and 2D NMR as well as HRMS spectra. In the case of **2**, the absolute configurations of their stereogenic carbons were established by comparison of their experimental and calculated electronic circular dichroism (ECD) spectra. Additionally, the absolute structures of tajixanthone hydrate (**6a**), and 15-acetyl tajixanthone hydrate (**6b**) were also unambiguously determined by X-ray analysis for the first time.

## 2. Results and Discussion

The structures of *p*-hydroxybenzaldehyde (**1**) [9], 1,3-dimethoxy-8-hydroxy-6-methylanthraquinone (**3a**) [10], 1,3-dimethoxy-2,8-dihydroxy-6-methylanthraquinone (**3b**) [11], emodin (**3c**) [12], 5[(3*E*,5*E*)-nona-3,5-dien-1-yl]benzene-1,3-diol (**5a**) [13], carnemycin E (**5b**) [13], tajixanthone hydrate (**6a**) [14,15], and 15-acetyl tajixanthone hydrate (**6b**) [16] were elucidated by the analysis of their 1D and 2D NMR spectra as well as HRMS data (Appendix A) and by the comparison of their NMR spectral data with those reported in the literature. 

Compound **2** was isolated as white crystals (mp 116–118 °C), and its molecular formula C_11_H_12_O_5_ was established based on the (+)-HRESIMS *m/z* 225.0765 [M + H]^+^ (calculated for C_11_H_13_O_5_, 225.0763) (Appendix A), requiring six degrees of unsaturation. The ^13^C NMR spectrum (Table 1, Appendix A), exhibited eleven carbon signals, which in combination with DEPT and HSQC spectra (Appendix A), can be categorized as one conjugated ester carbonyl (*δ*_C_ 169.1), two oxygen-bearing non-protonated sp^2^ (*δ*_C_ 166.2 and 164.5), two non-protonated sp^2^ (*δ*_C_ 142.1 and 100.0), two protonated sp^2^ (*δ*_C_ 106.8 and 101.3), two oxymethine sp^3^ (*δ*_C_ 77.8 and 67.5), one methoxy (*δ*_C_ 55.8) and one methyl (*δ*_C_ 15.9) carbon, respectively. The ^1^H NMR spectrum (Table 1, Appendix A), in combination with the HSQC spectrum (Appendix A), displayed a singlet of the hydrogen-bonded hydroxyl proton at *δ*_H_ 11.20 (OH-8), a multiplet of a hydroxyl proton at *δ*_H_ 2.28 (OH-4), two doublets of meta-coupled aromatic protons at *δ*_H_ 6.48 (*J* = 2.3 Hz, H-5/*δ*_C_ 106.8) and *δ*_H_ 6.45 (*J* = 2.3 Hz, H-7/*δ*_C_ 101.3), a double quartet at *δ*_H_ 4.63 (*J* = 6.6, 2 Hz, H-3/*δ*_C_ 77.8), which was coupled with a double doublet at *δ*_H_ 4.50 (*J* = 5.6, 1.5 Hz, H-4/*δ*_C_ 67.5), a methoxy singlet at *δ*_H_ 3.85 (*δ*_C_ 55.8) and a methyl doublet at *δ*_H_ 1.55 (*J* = 6.6 Hz, Me-9/*δ*_C_ 15.9). The HMBC spectrum (Table 1, Appendix A) exhibited correlations from OH-8 to the carbons at *δ*_C_ 164.5 (C-8), 101.3 (C-7) and 100.0 (C-8a), H-5 to the carbons at *δ*_C_ 166.2 (C-6), C-7, C-8a, C-4, H-7 to C-5, C-6, C-8a, H-3 to Me-9, H-4 to C-4a (*δ*_C_ 142.1), C-5, C-8a, OMe-6 to C-6, Me-9 to C-3, C-4, and a weak correlation from OH-4 to C-3 and C-4. 

The ^1^H and ^13^C NMR chemical shift values, together with COSY and HMBC correlations, revealed that the planar structure of **2** is the same as that of the enantiomeric mixture of (3*R*,4*R*)- and (3*S*,4*S*)-4-hydroxy-6-methoxymellein, previously isolated from the culture extract of the mycobionts of *Graphis* sp. Since the mixture showed a negative sign of Cotton effect at 267 nm, the authors proposed that the (3*R*,4*R*)-enantiomer was predominant [17]. Surprisingly, the NOESY spectrum (Table 1, Appendix A) of **2** exhibited a strong correlation from H-4 to Me-9, but not from H-3 to H-4, suggesting that H-4 and Me-9 are on the same face, which is contrary to (3*R*,4*R*)- and (3*S*,4*S*)-4-hydroxy-6-methoxymellein where H-3 and H-4 are on the same face. 

The absolute configurations of C-3 and C-4 were thus determined by comparison of the experimental ECD spectrum with a quantum-mechanically simulated spectrum derived from the most significant conformations of the computational models of (3*S*,4*R*)-**2** and (3*R*,4*S*)-**2** (Figure 2). Figure 3 shows a very good match between the experimental ECD spectrum and the calculated ECD spectrum for (3*S*,4*R*)-**2**, thus confirming that **2** is (3*S*,4*R*)-4-hydroxy-6-methoxymellein. Compound **2** is, therefore, a diastereomer of (3*R*,4*R*)- and (3*S*,4*S*)-4-hydroxy-6-methoxymellein [17] and has never been previously reported. 

Compound **4** was isolated as a red solid (mp. 228–229 °C), and its molecular formula C_23_H_19_NO_5_ was established on the basis of (+)-HRESIMS *m/z* 390.1340 [M + H]^+^ (calculated for C_23_H_20_NO_5_, 390.1341) (Appendix A), requiring fifteen degrees of unsaturation. The ^13^C NMR spectrum (Table 2, Appendix A) displayed 23 carbon signals which, in combination with DEPT and HSQC spectra (Appendix A), can be classified as two conjugated ketone carbonyls (*δ*_C_ 184.5 and 183.3), three oxygen-bearing non-protonated sp^2^ (*δ*_C_ 171.9, 161.2, 159.5), seven non-protonated sp^2^ (*δ*_C_ 146.7, 142.8, 134.4, 133.2, 123.2, 114.6, 100.0), seven protonated sp^2^ (*δ*_C_ 147.2, 146.2, 145.6, 127.5, 124.4, 120.4, and 118.6), two methylene sp^3^ (*δ*_C_ 33.8 and 23.7), and two methyl (*δ*_C_ 21.9 and 13.7) carbons. The ^1^H NMR spectrum (Table 2, Appendix A) showed two singlets of hydrogen-bonded phenolic hydroxyls at *δ*_H_ 12.50 and 13.44, seven aromatic protons appearing as a broad singlet at *δ*_H_ 8.91, two singlets at *δ*_H_ 7.11 and 6.80, three doublets at *δ*_H_ 8.57 (*J* = 8.0 Hz), 8.85 (*J* = 6.1 Hz), 7.46 (*J* = 1.0 Hz), one double doublet at *δ*_H_ 8.16 (*J* = 8.0, 6.1 Hz), one methylene triplet at *δ*_H_ 2.83 (*J* = 7.4 Hz), one methylene sextet at *δ*_H_ 1.70 (*J* = 7.4 Hz), one methyl singlet at *δ*_H_ 2.40, and one methyl triplet at *δ*_H_ 0.94 (*J* = 7.3 Hz). The presence of the 1,8-dihydroxy-6-methyl-1,2,3,6,8-pentasubstitututed anthraquinone scaffold was supported by COSY correlations (Table 2, Figure 4 and Appendix A) from H-7 (*δ*_H_ 7.11, s/*δ*_C_ 124.4) to H-5 (7.46, d, *J* = 1.0 Hz)/*δ*_C_ 120.4) and Me-11 (2.40, s/ *δ*_C_ 21.9), as well as by HMBC correlations (Table 2, Figure 4 and Appendix A) from OH-1 (*δ*_H_ 13.44, s) to C-1 (*δ*_C_ 159.5), C-2 (*δ*_C_ 123.2), C-9a (*δ*_C_ 100.0), OH-8 (*δ*_H_ 12.50, s) to C-7 (*δ*_C_ 124.4), C-8 (*δ*_C_ 161.2), C-8a (*δ*_C_ 114.6), H-5 (*δ*_H_ 7.46, d, *J* = 1.0 Hz) to C-10 (*δ*_C_ 183.3), C-7, C-8a, Me-11 (*δ*_C_ 21.9), H-7 (*δ*_H_ 7.11, s) to C-8, C-5 (*δ*_C_ 120.4), C-8a, Me-11, H-4 (*δ*_H_ 6.80,s) to C-10, C-2 (*δ*_C_ 123.2), and C-9a. That another portion of the molecule is 3-propylpyridinium was corroborated by COSY correlations (Table 2, Figure 4 and Appendix A) from H-4’ (*δ*_H_ 8.57, d, *J* = 8.0 Hz) to H-5’ (*δ*_H_ 8.16, dd, *J* = 8.0, 6.1 Hz) and H-2’ (*δ*_H_ 8.91, brs), H-5’ to H-4’ and H-6’ (8.85, d, *J* = 6.1 Hz), which was confirmed by HMBC correlations (Table 2, Figure 4 and Appendix A) from H-2’ to C-3’ (*δ*_C_ 142.8), C-4’ (*δ*_C_ 146.2) and C-1” (*δ*_C_ 33.8), H-4’ to C-2’ (*δ*_C_ 147.2), C-6’ (*δ*_C_ 145.6) and C-1”, H-5’ to C-3’, and C-6’ and H-6’ to C-4’ and C-5’ [18]. That the propyl group was on C-3’ of the pyridinium ring was supported by the spin system from H_2_-1” (*δ*_H_ 2.83, t, *J* = 7.4 Hz/*δ*_C_ 33.8) through H_2_-2” (*δ*_H_ 1.70, sex, *J* = 7.4 Hz/*δ*_C_ 23.7) to H_3_-3” (*δ*_H_ 0.94, t, *J* = 7.4 Hz/*δ*_C_ 13.7) as well as by HMBC correlations from H-3” to C-1” and C-2”, H-2” to C-1”, and C-3” and C-2’. Since H-2’ and H-6’ showed strong and weak cross peaks, respectively, to C-2 (Table 2, Figure 4 and Appendix A), the 3-propylpyridinium moiety is linked to the anthraquinone scaffold through the nitrogen atom for the former and C-2 of the latter. 

Since 1,8-dihydroxy-6-methyl anthraquinone and the 3-propylpyridinium moiety account for C_23_H_19_NO_4_, the only oxygen atom left must be on C-3 of the anthraquinone nucleus to produce a molecular formula C_23_H_19_NO_5_. Therefore, the oxygen atom on C-3 should bear a negative charge (**4**). This was supported by a high chemical shift value of C-3 (*δ*_C_ 171.9). This phenoxide ion can establish an ionic interaction with the positive-charged nitrogen of the pyridinium ring. Interestingly, although alkyl pyridinium-containing compounds have never been reported from marine-derived fungi, cyclic 3-alkylpyridinium alkaloids are common secondary metabolites from sponges of the order Haplosclerida [19,20]. Therefore, **4** is the first 3-alkylpyridinium anthraquinone reported from nature, and it was named stellatanthraquinone.

Compound **5b** was isolated as a pale-yellow viscous mass, and its molecular formula C_21_H_30_O_7_ was established on the basis of the (+)-HRESIMS *m/z* 395.2076 [M + H]^+^ (calculated for C_21_H_31_O_7_, 395.2070 (Appendix A), requiring seven degrees of unsaturation. Analysis of its ^1^H, ^13^C NMR, DEPT, COSY, HSQC, and HMBC spectra (Table 3, Appendix A) revealed the presence of a 5 [(3*E*,5*E*)-nona-3,5-dien-1-yl]benzene-1,3-diol moiety, identical to **5a**, with a substitution on C-2. The presence of five oxymethine sp^3^ (*δ*_C_ 81.5, 79.1, 75.0, 72.1, 70.3), one oxymethylene sp^3^ (*δ*_C_ 61.2) carbons, two hydroxyl groups at *δ*_H_ 4.90 dd (*J* = 10.7, 2.9 Hz) and 4.59 d (*J* = 5.5 Hz), and the molecular formula C_6_H_11_O_5_ of the substituent on C-2 revealed the presence of a pyranosyl moiety. However, since four oxymethine protons of the sugar moiety appeared as complex multiplets at *δ*_H_ 3.20–3.22 and 3.74, it was not possible to identify the sugar moiety of **5b**. Although Wen et al. [13] identified the sugar moiety in carnemycin E as glucopyranosyl, it was not possible to compare its ^1^H and ^13^C chemical shift values with those of the sugar moiety **5b** since the ^1^H and ^13^C NMR spectra of carnemycin E were obtained in pyridine-*d5*, while those of **5b** were obtained in DMSO-*d6*. Moreover, carnemycin E was obtained as an amorphous reddish gum, while **5b** was obtained as a pale-yellow viscous mass. In order to unravel the identity of the sugar moiety in **5b**, we tried to compare its carbon chemical shift values with those of the C-glycosides from the ^13^C NMR spectra obtained in DMSO-*d6*. The chemical shift values of C-1’, C-2’, C-3’, C-4’, C-5’ and C-6’ of the sugar moiety of **5b** (Table 3, Appendix A) were nearly identical to those of C-glucosyl moiety of tricetin 6,8-di-C-glucoside [21]. Moreover, the chemical shift value and coupling constant of H-1’ were also identical with those of the corresponding proton in tricetin 6,8-di-C-glucoside [21]. The value of the coupling constant of H-1’ (*J* = 9.6 Hz) confirmed the presence of a β-d-glucopyranosyl moiety. Therefore, **5a** was elucidated as carnemycin E, previously isolated from the culture extract of *Aspergillus* sp., which was isolated from superficial mycobiota of the brown alga, *Saccharina cichorioides* f. *sachalinensis,* collected from the South China Sea [13].

Compound **5c** was also isolated as a pale-yellow viscous mass and its molecular formula C_23_H_32_O_8_ was established on the basis of the (+)-HRESIMS *m/z* at 437.2175 [M + H]^+^ (calculated for C_23_H_33_O_8_, 437.2175), and *m/z* 459.1989 [M + Na]^+^ (calculated for C_23_H_32_O_8_Na, 459.1995) (Appendix A).The ^1^H and ^13^C NMR spectra of **5c** (Table 4; Appendix A) resembled those of **5b** (Appendix A) except for CH_2_-6’, which appeared at higher frequencies (*δ*_H_ 4.32 d*, J* = 11.6 Hz, and 3.98 dd, *J* = 11.6, 3.9 Hz/*δ*_C_ 64.8) than those of **5b** (*δ*_H_ 3.50, dd, *J* = 11.0, 5.5 Hz, and 3.65, dd*, J* = 11.0, 5.2 Hz)/*δ*_C_ 61.2) as well as the appearance of an acetyl group (*δ*_H_ 2.00, s/*δ*_C_ 21.2, CH_3_; *δ*_C_ 170.9, CO), suggesting that **5c** is a C-21 acetate of **5b**.

Contrary to other proton signals, the signals of OH-3, OH-3’, and OH-4’ appeared as broad signals in the ^1^H NMR spectrum at 500 MHz (Appendix A). Moreover, they did not show any COSY and HMBC correlations with any protons (Table 4, Appendix A), which made it impossible to assign them. Interestingly, in the ^1^H NMR spectrum at 300 MHz (Appendix A), the signal of OH-3 appeared as a sharp singlet at *δ*_H_ 8.70, whereas those of OH-3’ and OH-4’ appeared as two well-resolved doublets at *δ*_H_ 4.59, d *(J* = 6.5 Hz) and 5.15, d (*J* = 4.4 Hz), respectively. Furthermore, in the 300 MHz spectra, OH-3 displayed HMBC correlations to C-2 (*δ*_C_ 109.9) and C-3 (*δ*_C_ 157.2) (Appendix A), while OH-3’ and OH-4’ showed COSY correlations to H-3’ (*δ*_H_ 3.20) and H-4’ (*δ*_H_ 3.18) (Appendix A), respectively. The coupling constant of H-1’ (*J* = 9.8 Hz) confirmed the β-anomer of the glucosyl moiety. Since **5c** has never been previously reported, it was named acetyl carnemycin E

The ^1^H and ^13^C NMR spectra of **6a** and **6b** (Appendix A) are in agreement with those reported for tajixanthone hydrate [14] and 15-acetyl tajixanthone hydrate [16]. However, Pornpakakul et al. [14] assigned the configurations of C-15, C-20 and C-25 of tajixanthone hydrate, based on the coupling constant between H-14 and H-15 and NOESY correlations of the protons in tajixanthone methanoate, and also referred its stereochemistry to the previous study by Chexal et al. [22], who elegantly determined the absolute configurations of C-15 and C-25 of tajixanthone as 15*S* and 25*R* by chemical transformation (the method of Boar and Damps) while the relative configuration of C-20 was suggested by the preferred axial conformation of the isopropyl substituent in the hydrogenated derivatives [22]. Later on, the same group [23], described the isolation of tajixanthone hydrate which they obtained in a small quantity. The structure and stereochemistry of tajixanthone hydrate were identified on the basis of the same optical rotation of the acid-catalyzed hydrolysis product of tajixanthone and of the natural product. On the other hand, the absolute stereochemistry of 15-acetyl tajixanthone hydrate was concluded to be the same as that of tajixanthone hydrate, which was obtained by hydrolysis of 15-acetyl tajixanthone hydrate. However, neither optical rotation nor absolute configurations of their stereogenic carbons were provided [16]. 

Literature search revealed that the absolute configurations of C-15, C-20, and C-25 of both tajixanthone hydrate and 15-acetyl tajixanthone hydrate have never been established by either X-ray crystallographic or chiroptical methods. Fortunately, we were able to obtain suitable crystals of both **6a** and **6b** for X-ray analysis using an X-ray diffractometer equipped with CuKα radiation. The Ortep views of **6a** and **6b** are shown in Figure 5A and Figure 5B, respectively, revealing that the absolute configurations of C-15, C-20, and C-25 in both compounds are 15*S*, 20*S*, and 25*R*. Moreover, both compounds are levorotatory. 

The antimicrobial activity of **2**, **3a**, **3b**, **4**, **5a**, **5b**, **5c**, **6a**, and **6b** was evaluated against four reference bacterial species and three multidrug-resistant strains. However, only **5a** exhibited antibacterial activity against all Gram-positive strains, viz. *E. faecalis* ATCC 29212, vancomycin-resistant *E. faecalis* B3/101, and methicillin-resistant *S. aureus* 74/24, with a MIC value of 16 µg/mL, and *S. aureus* ATCC 29213, with a MIC value of 32 µg/mL (Table 5). The minimal bactericidal concentration (MBC) for **5a** was equal to or one-fold higher than the MIC, indicating its bactericidal effect towards *E. faecalis* ATCC 29212*, S. aureus* 74/24, and *S. aureus* ATCC 29213. For *E. faecalis* B3/101, its MBC was more than two-fold higher than the MIC, suggesting its bacteriostatic effect. 

Although **5a** was found to significantly inhibit NO production in lipopolysaccharide (LPS)-induced murine macrophage RAW264.7 cells [13], this compound has never been tested for antibacterial acivity. Interestingly, some alkenylresorcinols, such as 9-(3,5-dihydroxy-4-methylphenyl)nona-3(*Z*)-enoic acid, isolated from the methanolic extract of fruits of *Hakea sericea*, significantly inhibited the growth of *E. faecalis*, *Listeria monocytogenes* and *Bacillus cereus*, and showed good MIC values against *S. aureus* strains, including the clinical isolates and MRSA strains [24]. Intriguingly, **5b** and **5c**, analogs of **5a** which possess a β-glucosyl moiety on C-2 of the benzene ring, were void of antibacterial activity in our assays. We speculate that the polar and bulky glucosyl moiety might have negatively affected the antibacterial activity, possibly by preventing the compounds from penetrating the bacterial cell wall. 

Another interesting aspect is that even though there were several reports on the antibaterial activity of anthraquinones from marine-derived fungi [25], neither of the three anthraquinones tested, i.e., **3a**, **3b**, and **4**, showed antibacterial activity in our assays. This is not surprising since we also found in our previous report that the anthraquinone purnipurdin A, isolated from the culture extract of the marine sponge-associated fungus, *Neosartorya spinosa* KUFA 1047, did not exhibit any antibacterial activity against the same bacterial strains tested [26].

Compounds **2**, **3a**, **3b**, **4**, **5a**, **5b**, **5c**, **6a*,*** and **6b** were also investigated for their potential synergy with clinically relevant antibiotics on multidrug-resistant strains, by both disk diffusion method and checkerboard assay; however, no interactions were found with cefotaxime (ESBL *E. coli*), methicillin (MRSA *S. aureus*), and vancomycin (VRE *E. faecalis*).

The inhibitory effect of **2**, **3a**, **3b**, **4**, **5a**, **5b**, **5c**, **6a**, and **6b** on biofilm production was also evaluated in all reference strains. However, only **5a** showed an extensive ability to significantly inhibit biofilm formation in two of the four reference strains used in this study (Table 6). Indeed, **5a** was able to completely inhibit biofilm formation in *S. aureus* ATCC 29213 and *E. faecalis* ATCC 29212, at both MIC and 2xMIC concentrations. 

## 3. Experimental Sections

### 3.1. General Experimental Procedures

The melting points were determined on a Stuart Melting Point Apparatus SMP3 (Bibby Sterilin, Stone, Staffordshire, UK) and are uncorrected. Optical rotations were measured on an ADP410 Polarimeter (Bellingham + Stanley Ltd., Tunbridge Wells, Kent, UK). ^1^H and ^13^C NMR spectra were recorded at ambient temperature on a Bruker AMC instrument (Bruker Biosciences Corporation, Billerica, MA, USA) operating at 300 or 500 and 75 or 125 MHz, respectively. High resolution mass spectra were measured with a Waters Xevo QToF mass spectrometer (Waters Corporations, Milford, MA, USA) coupled to a Waters Aquity UPLC system. A Merck (Darmstadt, Germany) silica gel GF_254_ was used for preparative TLC, and a Merck Si gel 60 (0.2–0.5 mm) was used for column chromatography. LiChroprep silica gel and Sephadex LH 20 were used for column chromatography.

### 3.2. Fungal Material

The fungus was isolated from the marine sponge *Mycale* sp., which was collected by scuba diving at a depth of 5–10 m, from the coral reef at Samaesan Island (12°34′36.64″ N 100°56′59.69″ E), in the Gulf of Thailand, Chonburi province, in May 2015. The sponge was washed with 0.01% sodium hypochlorite solution for 1 min, followed by sterilized seawater three times, and then dried on a sterile filter paper under a sterile aseptic condition. The sponge was cut into small pieces (*ca*. 5 × 5 mm) and placed on Petri dish plates containing 15 mL potato dextrose agar (PDA) medium mixed with 300 mg/L of streptomycin sulfate and incubated at 28 °C for 7 days. The hyphal tips emerging from sponge pieces were individually transferred onto a PDA slant and maintained as pure cultures at Kasetsart University Fungal Collection, Department of Plant Pathology, Faculty of Agriculture, Kasetsart University, Bangkok, Thailand. The fungal strain KUFA 2017 was identified as *Aspergillus stellatus*, based on morphological characteristics such as colony growth rate and growth pattern on standard media, namely Czapek′s agar, Czapek yeast autolysate agar, and malt extract agar. Microscopic characteristics including size, shape, and ornamentation of conidiophores and spores were examined under light microscope. This identification was confirmed by molecular techniques using internal transcribed spacer (ITS) primers. DNA was extracted from young mycelia following a modified Murray and Thompson method [27]. Primer pairs ITS1 and ITS4 were used for ITS gene amplification [28]. PCR reactions were conducted on Thermal Cycler and the amplification process consisted of initial denaturation at 95 °C for 5 min, 34 cycles at 95 °C for 1 min (denaturation), at 55 °C for 1 min (annealing), and at 72 °C for 1.5 min (extension), followed by final extension at 72 °C for 10 min. PCR products were examined by Agarose gel electrophoresis (1% agarose with 1× Tris-Borate-EDTA (TBE) buffer) and visualized under UV light after staining with ethidium bromide. DNA sequencing analyses were performed using the dideoxyribonucleotide chain termination method [29] by Macrogen Inc. (Seoul, South Korea). The DNA sequences were edited using FinchTV software and submitted to the BLAST program for alignment and compared with that of fungal species in the NCBI database (http://www.ncbi.nlm.nih.gov/, accessed on 18 May 2021). Its gene sequences were deposited in GenBank with the accession number MZ331807.

### 3.3. Extraction and Isolation

The fungus was cultured in five Petri dishes (i.d. 90 mm) containing 20 mL of PDA per dish at 28 °C for one week. The mycelial plugs (5 mm in diameter) were transferred to two 500 mL Erlenmeyer flasks containing 200 mL of potato dextrose broth (PDB), and incubated on a rotary shaker at 120 rpm at 28 °C for one week. Thirty 1000 mL Erlenmeyer flasks, each containing 300 g of cooked rice, were autoclaved at 121 °C for 15 min. After cooling to room temperature, 20 mL of mycelial suspension of the fungus was inoculated per flask and incubated at 28 °C for 30 days, after which 500 mL of EtOAc was added to each flask of the moldy rice and macerated for 7 days, and then filtered with Whatman No. 1 filter paper. 

The EtOAc solutions were combined and concentrated under reduced pressure to yield 280.4 g of a crude EtOAc extract, which was dissolved in 300 mL of CHCl_3_, washed with H_2_O (3 × 500 mL) and dried with anhydrous Na_2_SO_4_, and filtered and evaporated under reduced pressure to obtain 134.9 g of a crude CHCl_3_ extract. The crude CHCl_3_ extract (57.1 g) was applied on a silica gel column (385 g) and eluted with mixtures of petrol-CHCl_3_ and CHCl_3_-Me_2_CO, wherein 250 mL fractions (frs) were collected as follows: frs 1–61 (petrol-CHCl_3_, 1:1), 62–129 (petrol-CHCl_3_, 3:7), 130–231 (petrol-CHCl_3_, 1:9), 232–397 (CHCl_3_-Me_2_CO, 9:1), 398–524 (CHCl_3_-Me_2_CO, 7:3). Frs 40–45 were combined (241.9 mg) and applied over a Sephadex LH-20 column (15 g), and eluted with MeOH, wherein 37 subfractions (sfrs) of 2 mL were collected. Sfrs 30–35 were combined (102.9 mg) and precipitated in MeOH to produce 13.6 mg of **3a**. Frs 74–77 were combined (851.6 mg) and precipitated in Me_2_CO to produce 20.3 mg of **3b**. Frs 78–89 were combined (396.5 mg) and applied over a Sephadex LH-20 column (15 g), and eluted with MeOH, wherein 28 sfrs of 2 mL were collected. Sfrs 21–25 were combined to produce 11.4 mg of **3c**. Frs 95–118 were combined (535.5 mg) and applied over a Sephadex LH-20 column (15 g), and eluted with MeOH, wherein 47 sfrs of 2 mL were collected. Sfrs 16–33 were combined (231.4 mg) and applied over another Sephadex LH-20 column (5 g), and eluted with CHCl_3_, wherein 22 sub-subfractions (ssfrs) of 1 mL were collected. Ssfrs 9–10 were combined to produce 69.4 mg of **2**, while ssfrs 20–22 were combined to produce 45.0 mg of **1**. Frs 126–140 were combined (274.9 mg) and applied over a Sephadex LH-20 column (15 g), and eluted with MeOH, wherein 31 sfrs of 2 mL were collected. Sfrs 4–15 were combined (108.4 mg) and precipitated in Me_2_CO to produce 26.7 mg of **6b**. Frs 149-173 were combined (1.02 g) and applied over a Sephadex LH-20 column (15 g), and eluted with MeOH, wherein 49 sfrs of 1 mL were collected. Sfrs 20–28 were combined (305.2 mg) and applied over another Sephadex LH-20 column (5 g), and eluted with CHCl_3_, wherein 20 ssfrs of 0.5 mL were collected. Ssfrs 15–19 were combined to produce 153.0 mg of **5a**. Frs 178–206 were combined (509.4 mg) and precipitated in MeOH to produce 46.8 mg of **6a**. Frs 237–238 were combined (209.3 mg) and applied over a Sephadex LH-20 column (15 g), and eluted with MeOH, wherein 20 sfrs of 2 mL were collected. Sfrs14-20 were combined (182.4 g) and applied over another Sephadex LH-20 column (5 g), and eluted with CHCl_3_, wherein 13 ssfrs of 1 mL were collected. Ssfrs 8–9 were combined to produce 7.3 mg of **5c**. Frs 437-455 were combined (323.3 mg) and applied over a Sephadex LH-20 column (15 g), and eluted with MeOH, wherein 24 sfrs of 2 mL were collected. Sfrs 11–15 were combined (211.9 mg) and applied over another Sephadex LH-20 column (5 g), and eluted with CHCl_3_, wherein 18 ssfrs of 0.5 mL were collected. Ssfrs 17–18 were combined to produce 141.7 mg of **5b**. Frs 460–513 were combined (297.3 mg) and applied over a Sephadex LH-20 column (15g), and eluted with MeOH, wherein 25 sfrs of 2 mL were collected. Sfrs 15–18 were combined (20.9 mg) and applied over another Sephadex LH-20 column (5 g), and eluted with CHCl_3_, wherein 13 ssfrs of 0.5 mL were collected. Ssfrs 4–7 were combined to produce 5.6 mg of **4**.

#### 3.3.1. (3*S*,4*R*)-4-Hydroxy-6-Methoxymellein (**2**)

White crystal. Mp 116–118 °C. [α${]}_{D}^{23}$ −200 (*c* 0.05, MeOH); For ^1^H and ^13^C spectroscopic data (CDCl_3_, 300 and 75 MHz), see Table 1; (+)-HRESIMS *m/z* 225.0765 [M + H]^+^ (calculated for C_11_H_13_O_5_, 225.0763).

#### 3.3.2. Stellatanthraquinone (**4**)

Red solid. Mp. 228–229 °C. For ^1^H and ^13^C spectroscopic data (DMSO-*d6*, 500 and 125 MHz), see Table 2; (+)-HRESIMS *m/z* 390.1340 [M + H]^+^ (calculated for C_23_H_20_NO_5_, 390.1341).

#### 3.3.3. Carnemycin E (**5b**) 

Pale-yellow viscous mass. [α]^20^_D_ +60 (*c* 0.05, MeOH). ^1^H and ^13^C spectroscopic data (DMSO-*d6*, 300 and 75 MHz), see Table 3; (+)-HRESIMS *m/z* 395.2076 [M + H]^+^ (calculated for C_21_H_331_O_7_, 395.2070).

#### 3.3.4. Acetyl Carnemycin E (**5c**)

Pale-yellow viscous mass. [α]^20^_D_ +260 (*c* 0.05, MeOH). ^1^H and ^13^C spectroscopic data (DMSO-*d6*, 500 and 125 MHz), see Table 4; (+)-HRESIMS *m/z* 437.2175 [M + H]^+^, (calculated for C_23_H_33_O_8_, 437.2175); *m/z* 459.1989 [M + Na]^+^ (calculated for C_23_H_32_O_8_Na, 459.1995).

### 3.4. X-ray Crystal Structures

Single crystals were mounted on cryoloops using paratone. X-ray diffraction data were collected at 290 K with a Gemini PX Ultra equipped with CuK_α_ radiation (λ = 1.54184 Å). The structures were solved by direct methods using SHELXS-97 and refined with SHELXL-97 [30]. Non-hydrogen atoms were refined anisotropically. Hydrogen atoms were either placed at their idealized positions using appropriate HFIX instructions in SHELXL and included in subsequent refinement cycles or were directly found from difference Fourier maps and were refined freely with isotropic displacement parameters.

#### 3.4.1. X-ray Crystal Structure of **6a**

Crystal was orthorhombic, space group *P*2_1_2_1_2_1_, cell volume 2178.1(4) Å^3^ and unit cell dimensions *a* = 6.2933(5) Å, *b* = 17.9862(18) Å and *c* = 19.243(3) Å (uncertainties in parentheses). Flack *x* [31] was refined parameter by means of TWIN and BASF in SHELXL to 0.0(5). The refinement converged to R (all data) = 8.88% and wR2 (all data) = 17.80%. Full details of the data collection and refinement and tables of atomic coordinates, bond lengths and angles, and torsion angles have been deposited with the Cambridge Crystallographic Data Centre (CCDC 2206108).

#### 3.4.2. X-ray Crystal Structure of **6b**

The crystal was orthorhombic, space group P2_1_2_1_2_1_, cell volume 2454.0(4) Å^3^, and unit cell dimensions *a* = 5.9772(6) Å, *b* = 13.8321(13) Å and *c* = 29.682(2) Å (uncertainties in parentheses). Calculated crystal density was 1.306 g/cm^−3^. The structure was solved by direct methods using SHELXS-97 and refined with SHELXL-97 [30]. The refinement converged to R (all data) = 8.45% and wR2 (all data) = 12.84% and Flack parameter = 0.0(3). Full details of the data collection and refinement and tables of atomic coordinates, bond lengths and angles, and torsion angles have been deposited with the Cambridge Crystallographic Data Centre (CCDC 2204631).

### 3.5. Electronic Circular Dichroism (ECD)

The experimental ECD spectrum of **2** (*ca.* 2 mg/mL in acetonitrile) was obtained in a Jasco J-815 CD spectropolarimeter (Jasco Europe S.R.L., Cremella, Italy) with a 0.1 mm cuvette and 6 accumulations. The simulated ECD spectra were obtained by first determining all the relevant conformers of the computational model. Its conformational space was developed by rotating all the single, non-ring, bonds for each of the two possible bends of the non-aromatic ring. The resulting 24 molecular mechanics conformers were minimized using the quantum mechanical DFT method B3LYP/6-31G with Gaussian 16W (Gaussian Inc., Wallingford, USA). The lowest 95% of the conformer Boltzmann populations (11 models) were subjected to a final minimization round using the method APFD/6-311+G(2d,p)/acetonitrile method (Gaussian 16W), which was also used, coupled with a TD method, to calculate its first 50 ECD transitions. The line spectrum for each one of the 11 models was built by applying a Gaussian line broadening of 0.15 eV to each computed transition with a constant UV-shift of 5 nm. The final ECD spectrum was obtained by the Boltzmann-weighted sum of the 11 line spectra [32].

### 3.6. Antibacterial Activity Bioassays

#### 3.6.1. Bacterial Strains and Testing Conditions

Four reference strains, obtained from the American Type Culture Collection (ATCC), viz. two Gram-positive (*Staphylococcus aureus* ATCC 29213, *Enterococcus faecalis* ATCC 29212), and two Gram-negative (*Escherichia coli* ATCC 25922, *Pseudomonas aeruginosa* ATCC 27853), were included in this study. Additionally, three multidrug-resistant strains including an extended-spectrum β-lactamase (ESBL)-producing *E. coli* (clinical isolate SA/2), and two environmental isolates, i.e., a methicillin-resistant isolate (MRSA) *S. aureus* 74/24 [33], and a vancomycin-resistant (VRE) isolate *E. faecalis* B3/101 [34]. All bacterial strains were cultured in MH agar (MH-BioKar Diagnostics, Allone, France) and incubated overnight at 37 °C before each assay. Stock solutions of each compound (4 mg/mL for the less soluble compounds, **3a** and **4**, and 10 mg/mL for the others) were prepared in dimethyl sulfoxide (DMSO-Alfa Aesar, Kandel, Germany), kept at −20 °C, and freshly diluted in the appropriate culture media before each assay. In all experiments, in-test concentrations of DMSO were kept below 1%, as recommended by the Clinical and Laboratory Standards Institute [35]. 

#### 3.6.2. Antimicrobial Susceptibility Testing

The Kirby–Bauer method was performed to screen the antimicrobial activity of the compounds according to CLSI recommendations [36]. Briefly, sterile blank paper discs with 6 mm diameter (Oxoid/Thermo Fisher Scientific, Basingstoke, UK) were impregnated with 15 µg of each compound and placed on MH plates, previously inoculated with a bacterial inoculum equal to 0.5 McFarland turbidity. After 18–20 h incubation at 37 °C, the diameter of the inhibition zones was measured in mm. Blank paper discs impregnated with DMSO were used as a negative control. Minimal inhibitory concentrations (MIC) were determined by the broth microdilution method, as recommended by the CLSI [37]. Two-fold serial dilutions of the compounds were prepared in cation-adjusted Mueller–Hinton broth (CAMHB-Sigma-Aldrich, St. Louis, MO, USA). With the exception of **3a** and **4**, the tested concentrations ranging from 1 to 64 µg/mL were used to keep in-test concentrations of DMSO below 1% to avoid bacterial growth inhibition. For **3a** and **4**, the highest concentration tested was 32 µg/mL. Colony forming unit (CFU) counts of the inoculum were conducted to ensure that the final inoculum size closely approximated the 5 × 10^5^ CFU/mL. The 96-well U-shaped untreated polystyrene plates were incubated for 16–20 h at 37 °C, and the MIC was determined as the lowest concentration of the compound that prevented visible growth. During the essays, vancomycin (VAN-Oxoid/Thermo Fisher Scientific, Basingstoke, UK) and oxacillin sodium salt monosulfate (OXA-Sigma-Aldrich, St. Louis, MO, USA) were used as positive controls for *E. faecalis* ATCC 29212 and *S. aureus* ATCC 29213, respectively. The minimal bactericidal concentration (MBC) was determined by spreading 10 µL of the content of the wells with no visible growth on MH plates. The MBC was defined as the lowest concentration to effectively reduce 99.9% of the bacterial growth after overnight incubation at 37 °C [38]. At least three independent assays were conducted for reference and multidrug-resistant strains.

#### 3.6.3. Antibiotic Synergy Testing

The Kirby–Bauer method was also used to evaluate the combined effect of the tested compounds with clinically relevant antibacterial drugs, as previously described [39]. A set of antibiotic discs (Oxoid/Thermo Fisher Scientific, Basingstoke, UK), to which the isolates were resistant, was selected: cefotaxime (CTX, 30 µg) for *E. coli* SA/2, vancomycin (VAN, 30 µg) for *E. faecalis* B3/101, and oxacillin (OXA, 1 µg) for *S. aureus* 74/24. Antibiotic discs impregnated with 15 µg of each compound were placed on seeded MH plates. The controls used included antibiotic discs alone, blank paper discs impregnated with 15 µg of each compound alone, and blank discs impregnated with DMSO. Plates with CTX were incubated for 18–20 h and plates with VAN and OXA were incubated for 24 h at 37 °C [35]. Potential synergy was considered when the inhibition halo of the antibiotic disc impregnated with compound was greater than the inhibition halo of the antibiotic or compound-impregnated blank disc alone.

The MIC method was also performed in order to evaluate the combined effect of the compounds and clinically relevant antimicrobial drugs. Briefly, when it was not possible to determine a MIC value for the tested compound, the MIC of CTX (Duchefa Biochemie, Haarlem, The Netherlands), VAN (Oxoid, Basingstoke, England), and OXA (Sigma-Aldrich, St. Louis, MO, USA) for the respective multidrug-resistant strains was determined in the presence of the highest concentration of each compound tested in previous assays (64 µg/mL or 32 µg/mL for compounds **3a** and **4**). The tested antibiotic was serially diluted whereas the concentration of each compound was kept fixed. Antibiotic MICs were determined as described above. Potential synergy was considered when the antibiotic MIC was lower in the presence of compound [40]. Fractional inhibitory concentrations (FIC) were calculated as follows: FIC of the compound = MIC of the compound combined with antibiotic/MIC of the compound alone, and FIC antibiotic = MIC of antibiotic combined with compound/MIC of antibiotic alone. The FIC index (FICI) was calculated as the sum of each FIC and interpreted as follows: FICI ≤ 0.5, ‘synergy’; 0.5 < FICI ≤ 4, ‘no interaction’; 4 < FICI, ‘antagonism’ [41]. 

#### 3.6.4. Biofilm Formation Inhibition Assay

In order to evaluate the antibiofilm activity of the compounds, the crystal violet method was used to quantify the total biomass produced [39,42]. Briefly, the highest concentration of the compound tested in the MIC assay was added to bacterial suspensions of 1 × 10^6^ CFU/mL prepared in unsupplemented Tryptone Soy broth (TSB-Biokar Diagnostics, Allone, Beauvais, France) or TSB supplemented with 1% (*w/v*) glucose (d-(+)-glucose anhydrous for molecular biology, PanReac AppliChem, Barcelona, Spain) for Gram-positive strains. When it was possible to determine the MIC, concentrations between 2× MIC and ¼ MIC were tested, while keeping in-test concentrations of DMSO below 1%. When it was not possible to determine the MIC value, the concentration tested was 64 µg/mL (or 32 µg/mL for compounds **3a** and **4**). Controls with appropriate concentration of DMSO, as well as a negative control (TSB or TSB + 1% glucose alone) were included. Sterile 96-well flat-bottomed untreated polystyrene microtiter plates were used. After a 24 h incubation at 37 °C, the biofilms were heat-fixed for 1 h at 60 °C and stained with 0.5% (*v/v*) crystal violet (Química Clínica Aplicada, Amposta, Spain) for 5 min. The stain was resolubilized with 33% (*v/v*) acetic acid (Acetic acid 100%, AppliChem, Darmstadt, Germany) and the biofilm biomass was quantified by measuring the absorbance of each sample at 570 nm in a microplate reader (Thermo Scientific Multiskan^®^ FC, Thermo Fisher Scientific, Waltham, MA, USA). The background absorbance (TSB or TSB + 1% glucose without inoculum) was subtracted from the absorbance of each sample and the data are presented as percentage of control. Three independent assays were performed for reference strains, with triplicates for each experimental condition.

## 4. Conclusions

The EtOAc extract of the culture of a marine-derived fungus, *Aspergillus stellatus* KUFA 2017, isolated from a marine sponge *Mycale* sp., which was collected in the Gulf of Thailand, furnished three previously unreported secondary metabolites viz. (3*S*,4*R*)-4-hydroxy-6-methoxymellein (**2**), a structurally unique propylpyridinium anthraquinone, stellatanthraquinone (**4**), and acetyl carnemycin E (**5c**), in addition to eight previously reported compounds, including *p*-hydroxybenzaldehyde (**1**), 1,3-dimethoxy-8-hydroxy-6-methylanthraquinone (**3a**), 1,3-dimethoxy-2,8-dihydroxy-6-methylanthraquinone (**3b**), emodin (**3c**), 5[(3*E*,5*E*)-nona-3,5-dien-1-yl]benzene (**5a**), carnemycin E (**5b**), tajixanthone hydrate (**6a**), and 15-acetyl tajixanthone hydrate (**6b**). While the absolute configurations of the stereogenic carbons in **2** were established unambiguously by a combination of NOESY correlations and a comparison of experimental and calculated ECD spectra, the stereostructures of **6a** and **6b** were established by X-ray analysis for the first time. 

All the compounds, except **1** and **3c**, were evaluated for their antibacterial activity against four reference strains: two Gram-positive (*Staphylococcus aureus* ATCC 29213, *Enterococcus faecalis* ATCC 29212) and two Gram-negative (*Escherichia coli* ATCC 25922, *Pseudomonas aeruginosa* ATCC 27853), as well as three multidrug-resistant strains including an extended-spectrum β-lactamase (ESBL)-producing *E. coli* (clinical isolate SA/2), a methicillin-resistant isolate (MRSA) *S. aureus* 74/24 and a vancomycin-resistant (VRE) isolate *E. faecalis* B3/101. However, only **5c** exhibited antibacterial activity against all Gram-positive strains with a MIC value of 16 µg/mL toward *E. faecalis* ATCC 29212, vancomycin-resistant *E. faecalis* B3/101, and methicillin-resistant *S. aureus* 74/24, but with a higher MIC value (32 µg/mL) toward *S. aureus* ATCC 29213. Since **5a** displayed the minimal bactericidal concentration (MBC) equal to or one-fold higher than the MIC, it was suggested that **5a** exerted a bactericidal effect towards *E. faecalis* ATCC 29212*, S. aureus* 74/24, and *S. aureus* ATCC 29213. On the contrary, the MBC of **5a** was more than two-fold higher than the MIC toward *E. faecalis* B3/101; therefore, this compound was suggested to have a bacteriostatic effect against this multidrug-resistant species. Interestingly, **5b** and **5c**, which are C-glucosylated **5a**, were void of antibacterial activity against all the tested organisms. These results lead to a conclusion that the polar and bulky glucosyl substituent on the benzene ring can negatively affect the antibacterial activity of this series of compounds. Finally, **5a** was also able to completely inhibit biofilm formation in *S. aureus* ATCC 29213 and *E. faecalis* ATCC 29212 at both MIC and 2× MIC concentrations. Since **5a** possesses interesting antibacterial and potent antibiofilm activities, this compound can be considered as an interesting model for the development of a new type of antibiotics.

## Figures and Tables

**Figure 1 marinedrugs-20-00672-f001:**
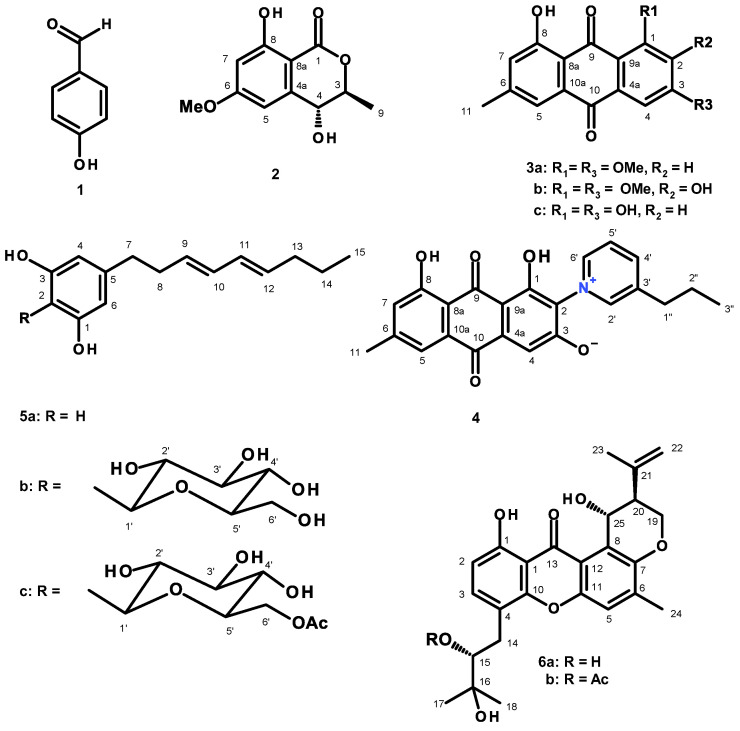
Structures of *p*-hydroxybenzaldehyde (**1**), (3*S*,4*R*)-4-hydroxy-6-methoxymellein (**2**), 1,3-dimethoxy-8-hydroxy-6-methylanthraquinone (**3a**), 1,3-dimethoxy-2,8-dihydroxy-6-methylanthraquinone (**3b**), emodin (**3c**), stellatanthraquinone (**4**), 5[(3*E*,5*E*)-nona-3,5-dien-1-yl]benzene-1,3-diol (**5a**), carnemycin E (**5b**), acetyl carnemycin E (**5c**), tajixanthone hydrate (**6a**), 15-acetyl tajixanthone hydrate (**6b**).

**Figure 2 marinedrugs-20-00672-f002:**
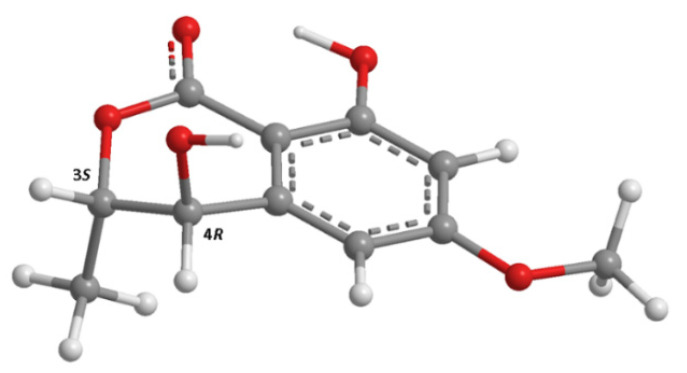
Model of the most abundant conformation of **2** (lowest APFD/6-311+G(2d,p)/acetonitrile energy conformer), accounting for 25% of conformer population, in its ECD-assigned (3*S*,4*R*) configuration.

**Figure 3 marinedrugs-20-00672-f003:**
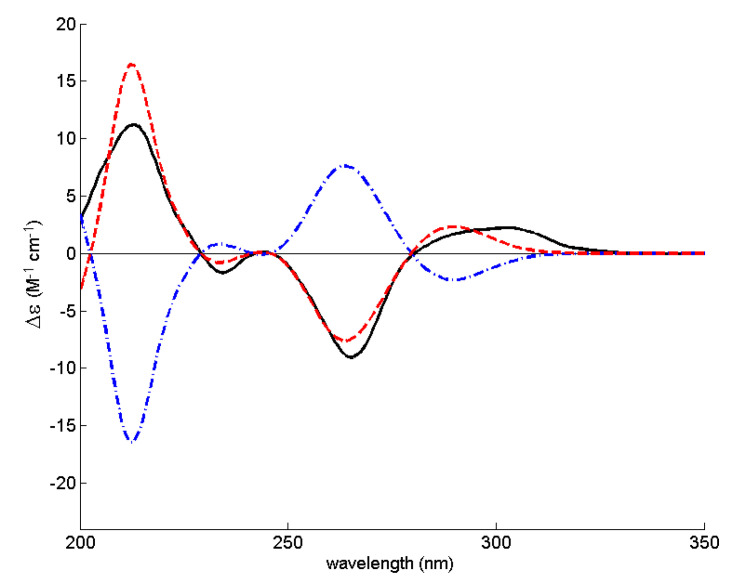
Experimental acetonitrile ECD spectrum of **2** (solid black line) and theoretical ECD spectra of its (3*R*,4*S*) (dot-dashed blue line) and (3*S*,4*R*) (dashed red line) computational models. Of the two theoretical spectra, the (3*R*,4*S*) was the one actually simulated and the (3*S*4,*R*) was obtained by changing the sign of every point of the (3*R*,4*S*) spectrum.

**Figure 4 marinedrugs-20-00672-f004:**
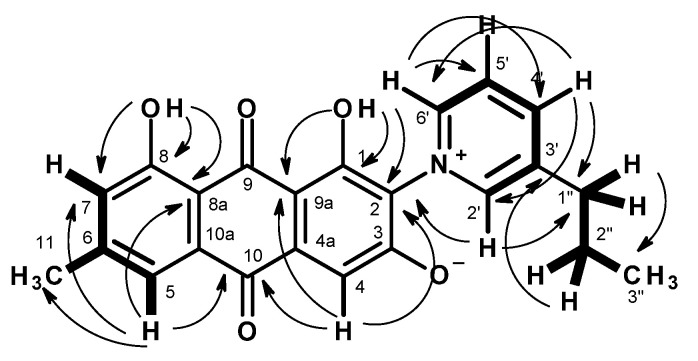
Key COSY (

) and HMBC (

) correlations in **4**.

**Figure 5 marinedrugs-20-00672-f005:**
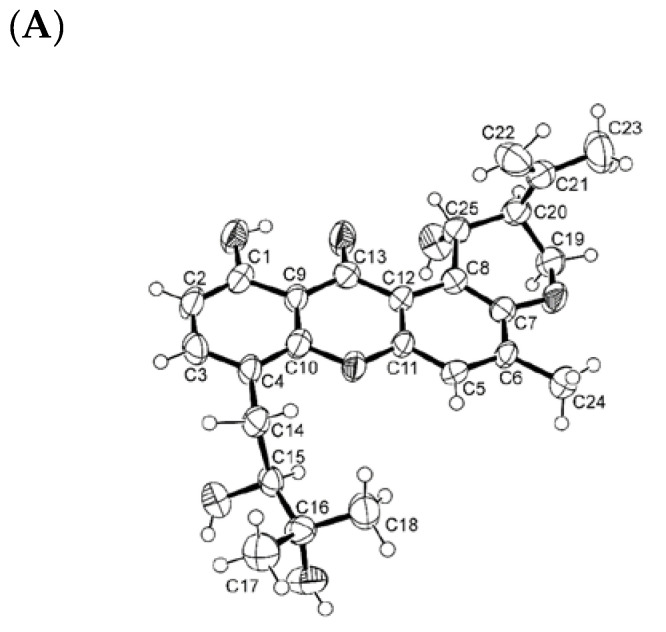
Ortep views of **6a** (**A**) and **6b** (**B**).

**Table 1 marinedrugs-20-00672-t001:** ^1^H and ^13^C NMR data (CDCl_3_, 300 and 75 MHz), COSY, HMBC, and NOESY for **2**.

Position	*δ*_C_, Type	*δ*_H_, (*J* in Hz)	COSY	HMBC	NOESY
1	169.1, CO	-			
3	77.8, CH	4.63, dq (6.6, 2.0)	CH_3_-9	CH_3_-9	CH_3_-9
4	67.5, CH	4.50, dd (5.6, 1.5)	OH-4	C-4a, 5, 8a	H-5, CH_3_-9
4a	142.1, C	-	-		
5	106.8, CH	6.48, d (2.3)	H-7	C-4, 6, 7, 8a	H-4, OCH_3_-6
6	166.2, C	-			
7	101.3, CH	6.45, d (2.3)	H-5,	C-5, 6, 8a	OCH_3_-6
8	164.5, C	-			
8a	100.0, C	-			
9	15.9, CH_3_	1.55, d (1.6)	H-3	C-3, 4	H-3, 4
OCH_3_-6	55.8, CH_3_	3.85, s		C-6	
OH-4	-	2.28, m		C-3, 4	
OH-8	-	11.20, s		C-7, 8, 8a	

**Table 2 marinedrugs-20-00672-t002:** ^1^H and ^13^C NMR data (DMSO-*d_6_*, 500 and 125 MHz), COSY and HMBC assignment for **4**.

Position	*δ*_C_, Type	*δ*_H_, (*J* in Hz)	COSY	HMBC
1	159.5, C			
2	123.2, C			
3	171.9, C			
4	118.6, CH	6.80, s		C-2, 9a, 10
4a	133.2, C			
5	120.4, CH	7.46, d (1.0)	H-7	C-7, 8a, 10, CH_3_-11
6	146.7, C			
7	124.4, CH	7.11, s	H-5, CH_3_-11	C-5, 8, 8a, CH_3_-11
8	161.2, C			
8a	114.6, C			
9	184.5, CO			
9a	100.0, C			
10	183.3, CO			
10a	134.4, C			
11	21.9, CH_3_	2.40, s	H-7	C-5, 6,7
2’	147.2, CH	8.91, brs	H-4’	C-2, 3’, 4’, 1”
3’	142.8, C			
4’	146.2, CH	8.57, d (8.0)	H-2’,5’	C-2’, 6’, 1”
5’	127.5, CH	8.16, dd (8.0, 6.1)	H-4’, 6’	C-3’, 6’
6’	145.6, CH	8.85, d (6.1)	H-5’	C-2, 4’, 5’
1”	33.8, CH_2_	2.83, t (7.4)	H-2”	C-2’, 2”, 3’, 3”
2”	23.7, CH_2_	1.70, sex (7.4)	H-1”, 3”	C-1”, 3’, 3”
3”	13.7, CH_3_	0.94, t (7.3)	H-2”	C-1”, 3”
OH-1		13.44, s		C-1, 2, 9a
OH-8		12.50, s		C-7, 8, 8a

**Table 3 marinedrugs-20-00672-t003:** ^1^H and ^13^C NMR data (DMSO-*d_6_*, 300 and 75 MHz), COSY, and HMBC assignment for **5b**.

Position	*δ*_C_, Type	*δ*_H_, (*J* in Hz)	COSY	HMBC
1	157.1, C			
2	110.2, C			
3	157.1, C			
4	107.6, CH	6.11, s		C-1’, 3, 6, 7
5	142.5, C			
6	107.6,CH	6.11, s		C-1’, 2, 4, 7
7	35.5, CH_2_	2.45, t (6.8)	H-8	C-4, 5, 6, 8, 9
8	34.0, CH_2_	2.27, dd (14.3, 6.6)	H-7, H-9	C-5, 7, 9, 10
9	131.8, CH	5.62, m	H-8, 10	C-10, 11
10	130.9, CH	5.91, m	H-9, 11	C-8, 9, 12
11	131.1, CH	6.04, m	H-10, 12	C-9, 13
12	132.6, CH	5.55, m	H-11, 13	C-10, 13, 14
13	34.5, CH_2_	2.00, dd (14.3, 7.2)	H-12, 14	C-11, 12, 14, 15
14	22.5, CH_2_	1.36, sext (7.2)	H-13, 15	C-12, 13, 15
15	14.0, CH_3_	0.87, t (7.2)	H-14	C-13, 14
1’	75.0, CH	4.62, d (9.6)	H-2’	C-1, 2, 2’, 3, 3’
2’	72.1, CH	3.74, m		
3’	79.1, CH	3.22, m		
4’	70.3, CH	3.22, m	OH-4’	
5’	81.5, CH	3.20, m		
6’a b	61.2, CH_2_	3.50, dd (11.0, 5.5) 3.65, dd (11.0, 5.2)	H-5’, 6’b, OH-6’ H-5’, 6’a, OH-6’	
OH-3		8.67, s		C-2, 3, 4
OH-4’		4.90, dd (10.7, 2.9)	H-4’	C-4’, 5’
OH-6’		4.59, d (5.5)	H_2_-6’	C-6’

**Table 4 marinedrugs-20-00672-t004:** ^1^H and ^13^C NMR data (DMSO-*d_6_*, 500 and 125 MHz), COSY, and HMBC assignment for **5c**.

Position	*δ*_C_, Type	*δ*_H_, (*J* in Hz)	COSY	HMBC
1	157.3, C			
2	109.9, C			
3	157.3, C			
4	107.5, CH	6.11, s		C-1’, 2, 3, 6, 7
5	142.5, C			
6	107.5, CH	6.11, s		C-1’, 2, 3, 4, 7
7	35.5, CH_2_	2.44, t (7.2)	H-8	C-4, 5, 6, 8, 9
8	34.0, CH_2_	2.26, dd (14.7, 7.2)	H-7, 9	C-7, 9, 10
9	131.8, CH	5.59, ddd (14.6, 7.2, 7.2)	H-8, 10	C-11
10	130.9, CH	5.97, m	H-9, 11	
11	131.1, CH	6.04, m	H-10, 12	C-9 (w), 13
12	132.6, CH	5.57, ddd (14.5, 7.7, 7.1)	H-11, 13	C-10, 13, 14
13	34.5, CH_2_	2.01, m	H-12, 14	C-11, 12, 14, 15
14	22.5, CH_2_	1.36, sex (7.4)	H-13, 15	12, 13, 15
15	14.0, CH_3_	0.86, t (7.4)	H-14	C-13, 14
1’	74.9, CH	4.60, d (9.8)	H-2’	C-1, 2, 2’ 3, 3’
2’	71.5, CH	3.83, t (9.2)	H-1’, 3’	C-1’, 3’
3’	79.0, CH	3.21, t (8.7)	H-4’	C-4’
4’	70.6, CH	3.18, t (8.7)	H-3’	C-3’
5’	78.4, CH	3.36 (under water peak)	H-6’b	C-4’ (w)
6’a b	64.8, CH_2_	4.32, d (11.6) 3.98 dd (11.6, 3.9)	H-6’b H-5’, 6’a	C-4’, CO (OAc) C-5’, 6’
OAc	170.9, CO	-		
OAc	21.2, CH_3_	2.00, s		C-6’
OH-3		8.74, brs		
OH-3’		4.95, br		
OH-4’		5.15, br		

**Table 5 marinedrugs-20-00672-t005:** Antibacterial activity of **2**, **3a**, **3b**, **4**, **5a**, **5b**, **5c**, **6a**, and **6b** against Gram-positive reference and multidrug-resistant strains. MIC and MBC are expressed in µg/mL.

Compound	*E. faecalis* ATCC 29212		*E. faecalis* B3/101 (VRE)		*S. aureus* ATCC 29213		*S. aureus* 74/24 (MRSA)	
	MIC	MBC	MIC	MBC	MIC	MBC	MIC	MBC
**2**	>64	>64	>64	>64	>64	>64	>64	>64
**3a**	>32	>32	>32	>32	>32	>32	>32	>32
**3b**	>64	>64	>64	>64	>64	>64	>64	>64
**4**	>32	>32	>32	>32	>32	>32	>32	>32
**5a**	16	32	16	64	32	32	16	32
**5b**	>64	>64	>64	>64	>64	>64	>64	>64
**5c**	>64	>64	>64	>64	>64	>64	>64	>64
**6a**	>64	>64	>64	>64	>64	>64	>64	>64
**6b**	>32	>32	>32	>32	>32	>32	>32	>32
VAN	4	-	-	-	-	-	-	-
OXA	-	-	-	-	0.2	-	-	-

MIC, minimal inhibitory concentration; MBC, minimal bactericidal concentration; VAN, vancomycin; OXA, oxacillin.

**Table 6 marinedrugs-20-00672-t006:** Percentage of biofilm formation in the presence of **5a** after 24 h incubation.

Compound	Concentration	Biofilm Biomass (% of Control)	
		*E. faecalis* ATCC 29212	*S. aureus* ATCC 29213
**5a**	64 µg/mL	-	0.00 *±* 0.06 ***
	32 µg/mL	0.01 *±* 0.01 ***	0.04 *±* 0.12 ***
	16 µg/mL	0.02 *±* 0.01 ***	-
DMSO	1% (*v/v*)	1.00 *±* 0.03 ***	1.00 *±* 0.01 ***

Data are shown as mean *±* SD of three independent experiments. One-sample *t*-test: * *p* < 0.05, ** *p* < 0.01, *** *p* < 0.001 significantly different from 100%.

## Data Availability

Data sharing is not applicable.

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
