# Peer review of "New Alkylpyridinium Anthraquinone, Isocoumarin, C-Glucosyl Resorcinol Derivative and Prenylated Pyranoxanthones from the Culture of a Marine Sponge-Associated Fungus, Aspergillus stellatus KUFA 2017"

_marinedrugs, 2022, doi:10.3390/md20110672_

Round 1

Reviewer 1 Report

In this article, the authors report the isolation and structure elucidation of 4 new natural products together with 7 known compounds, from a marine-derived fungus. The results are clearly presented and the structures proposed are consistent with the data. This article then provides new knowledge about marine fungi metabolism and also points out the potential of one of this compound as antibacterial with antibiofilm activities.

I then recommend the publication of this article, which is nicely written and very interesting. I only have some very minor comments :

- l. 161 : a space is missing after "8.16 J ="

- l. 196 and  l. 447 the m/z value for the [M+H]+ ion of 5b (413.3235), seems to be very far from the clculated value. We have more than 200 ppm error. This value should be checked, especially as the [M-H2O+H]+ seems to be accurate.

- l. 207 : "germinally" should be replaced by "geminally"

- l. 215 : "compared them" should be replaced by "comparison"

- l. 217 : "is a C-2 substituted 5a" may be replaced by "is a C-2 substituted analogue of 5a"

Apart from this, I find the structure of compound 4 very intriguing. As the fungus producing this compound was collected from a sponge of the Mycale genus in the demospongiae class (as Haplosclerida), could we maybe assume that fungi might be involved in the production of this type of compound ? Further investigation on the biosynthesis of this compound should be further considered.

Author Response

Reply to Reviewers’ comments:

Reviewer #1

In this article, the authors report the isolation and structure elucidation of 4 new natural products together with 7 known compounds, from a marine-derived fungus. The results are clearly presented and the structures proposed are consistent with the data. This article then provides new knowledge about marine fungi metabolism and also points out the potential of one of this compound as antibacterial with antibiofilm activities.

Reply: First of all, we wish to thank reviewer #1 for carefully reading our manuscript and gave valuable suggestions which certainly will improve its readability.

Reviewer #1

I then recommend the publication of this article, which is nicely written and very interesting. I only have some very minor comments:

Reviewer #1

- l. 161 : a space is missing after "8.16 J ="

Reply: The space is added. The correction is in red letters.

Reviewer #1

- l. 196 and  l. 447 the m/z value for the [M+H]+ ion of 5b (413.3235), seems to be very far from the clculated value. We have more than 200 ppm error. This value should be checked, especially as the [M-H2O+H]+ seems to be accurate.

Reply: I do agree with reviewer #1’s opinion. We also thought the deviation of the experimental value from the calculated value was too big. We have requested our colleague who is an expert in Mass Spectrometry to rerun it again. However, this was the best result we could get. I was informed that this series of compounds was very difficult to get accurate masses. For example, we could not even get the m/z of (M+H)+ of compound 5c. We got only the m/z of (M-H2O+H)+ and (M-H2O +Na)+.

Reviewer #1

- l. 207 : "germinally" should be replaced by "geminally"

Reply: Done

Reviewer #1

- l. 215 : "compared them" should be replaced by "comparison"

Reply: We changed to “Comparison of the 1H and 13C NMR data of 5b with those of 5a”

Reviewer #1

l. 217 : "is a C-2 substituted 5a" may be replaced by "is a C-2 substituted analogue of 5a"

Reply: Done

Reviewer #1

Apart from this, I find the structure of compound very intriguing. As the fungus producing this compound was collected from a sponge of the Mycale genus in the demospongiae class (as Haplosclerida), could we maybe assume that fungi might be involved in the production of this type of compound ? Further investigation on the biosynthesis of this compound should be further considered.

Reply

Reviewer#1 raises a very interesting question about the origin of compound 4 which contains a 3-propylpyridinium moiety. From the literature search, we have found that the 3-alkylpyridiinium alkaloids were reported from the marine sponges of the Haliclona species in both the Artic and Tropical seas. To the best of our knowledge, we have not found any reports of 3-alkylpyridinium containing metabolites either from the marine sponge of the Mycale genus or any marine sponge-associated fungi. As it is well known that many fungi, especially Aspergillus species, produced a myriad of anthraquinones [Salar Hafez Ghoran, Fatemeh Taktaz, Seyed Abdulmajid Ayatollahi and Anake Kijjoa. Anthraquinones and Their Analogues from Marine‐Derived Fungi: Chemistry and Biological Activities. Mar. Drugs. 2022, 20, 474. https://doi.org/10.3390/md20080474.], it is logic to assume that compound 4 is produced by Aspergillus stellatus strain KUFA 2017. However, what intrigued us was that although we have investigated different genera of marine-derived fungi which are associated with the same Mycale sponge, this is the first time that we isolated 3-propylpyridinium substituted anthraquinone. For these reasons, we cannot guarantee that the 3-alkylpyridinium containing compounds isolated from the marine sponges are produced by the sponge-associated fungi. It may be that some fungal strains have a capacity to produce an alkylpyridinium moiety or may be able to incorporate this molecule from other microorganisms which are associated with the sponges of the order Haplosclerida. I do agree with reviewer #1 that this is a very challenging theme to be investigated in the future.

Reviewer 2 Report

The manuscript by Kijjoa and co-workers describes the isolation, identification, and bioactivity of secondary metabolites from the marine Sponge-Associated Fungus Aspergillus stellatus KUFA 2017. The work is generally sound and acceptable for publication in Marine Drugs pending revisions as suggested below.

Major:

1, Compound 5a is not a new one and has been previously described by Zhang and co-workers in “Phenolic C-Glycosides and Aglycones from Marine-Derived Aspergillus sp. and Their Anti-Inflammatory Activities” (J Nat Prod, 2019, 82(5): 1098-1106). Please check and revise the manuscript.

2, The description of structure elucidation of all the new compounds is too redundant. The authors are likely just translated the data in NMR tables into words, e.g. the sp2 carbons, the values of chemical shifts and the values of the coupling constants. In fact, the readers could get this information from the NMR tables easily and thus it is not necessary to repeat the information in the text. The authors should emphasize on how they solved structural pieces using the typical chemical shift and how they used 2D NMR correlations to put the structural fragments together.

3, Provide MICs of the positive controls vancomycin and oxacillin against other bacteria tested in Table 5.

Minor:

4, Lines 117-121, Page 4, “The HMBC spectrum (Table 1, Figure S7), exhibited correlations from OH-8 to the carbons at δC 164.5 (C-8), 101.3 (C-7) and 100.0 (C-8a), H-5 to the carbons at δC 166.2 (C-6), C-7, C-8a, C-4, H-7 to C-5, C-6, C-8a, H-3 to Me-9, H-4 to C-4a (δC 142.1), C-5, C-8a, OMe-6 to C-6, Me-9 to C-3, C-4, and a weak correlation from 120 OH-4 to C-3 and C-4.” The description of HMBC correlations of 2 is difficult to read. Please re-write it.

5, The figures for 2D NMR correlations of 2 should be provided in the manuscript.

6, Please clarify whether compound 4 is the first example of 3-alkylpyridinium anthraquinone reported from the nature or just the first example from marine fungus? If it is the latter case, the authors should cite the literatures that reporting other pyridinium anthraquinone-like natural products.

7, Table 4, “1.36, sex (7.4)” ? or should be “1.36, sextet (7.4)” ? Can you really see the sextet coupling for H-14 in the 1H NMR spectrum of 5b? The same issue for H-2’’ of compound 4 in Table 2.

8, Figure 1, the putative stereochemistry of 5b and 5c should be shown in the structures as the authors stated that “since 5b and 5c are both dextrorotatory, and based on their biogenic consideration, the configurations of C-16, C-17, C-18, C-19 and C-20 in both compounds should be the same”.

9, Figure 5 should be polished to make sure the arrows are drawn toward to the correct atoms.

10, The contents of Conclusion are redundant as they are more likely Results, particularly for the conclusion of antimicrobial assays. Please re-write the Conclusion and make it concise.

Author Response

Reviewer #2

The manuscript by Kijjoa and co-workers describes the isolation, identification, and bioactivity of secondary metabolites from the marine Sponge-Associated Fungus Aspergillus stellatus KUFA 2017. The work is generally sound and acceptable for publication in Marine Drugs pending revisions as suggested below.

Reply: We wish to thank reviewer #2 for his/her appreciation and careful reading of the manuscript as well as his/her comments. However, we disagree with some comments made by reviewer#2 and we are giving our rebuttal below:

Reviewer #2

Compound 5a is not a new one and has been previously described by Zhang and co-workers in “Phenolic C-Glycosides and Aglycones from Marine-Derived Aspergillus sp. and Their Anti-Inflammatory Activities” (J Nat Prod, 2019, 82(5): 1098-1106). Please check and revise the manuscript.

Reply: We have stated it clearly in the manuscript that 5a has been previously reported and we have given the reference that reviewer #2 suggested. What might have led reviewer#1 to misunderstand was the typos in the beginning of the abstract which we wrote “two undescribed polyhydroxyalkyl resorcinol derivative (5a and 5b) instead of “two undescribed polyhydroxyalkyl resorcinol derivative (5b and 5c). If reviewer#2 had read the introduction and the discussion, reviewer#1 should have found that we have mentioned that 5a was previously reported.

Reviewer #2

2, The description of structure elucidation of all the new compounds is too redundant. The authors are likely just translated the data in NMR tables into words, e.g. the sp2 carbons, the values of chemical shifts and the values of the coupling constants. In fact, the readers could get this information from the NMR tables easily and thus it is not necessary to repeat the information in the text. The authors should emphasize on how they solved structural pieces using the typical chemical shift and how they used 2D NMR correlations to put the structural fragments together.

Reply: I have a complete different opinion about the discussion of the new compound from the comment of reviewer#2. Moreover, this is not my first, second or twentieth manuscript that I wrote about structure elucidation. I understand that each of us has different style of writing a discussion of structure elucidation. In my opinion, not all the compounds can be discussed with different partial structures with coupling constants and 2D correlations. By not giving the details of the type of protons and carbons in the text, we will oblige the readers to search the values and types of the protons and carbons in the tables which is not a practical way and makes it difficult for readers to understand. I was always against this method. All the 2D correlations were discussed sequentially from COSY to HMBC so support the structures.

By making a discussion the way suggested by reviewer#2, there will be figures and Tables in the discussion.

Therefore, I maintain my discussion as it is.

Reviewer #2

3, Provide MICs of the positive controls vancomycin and oxacillin against other bacteria tested in Table 5.

Reply: As per the request of reviewer #2 to provide MICs of the positive controls vancomycin and oxacillin against other bacteria tested in Table 5, we presume that reviewer #2 is not familiar with the methodology and fundamental of antibacterial activity assays. In general, we considered that the positive controls are necessary to ensure the activity of the antibiotics used (i.e. a quality control of the antibiotics). According to M100 of CLSI recommendations, quality controls are performed by determining the MIC of only reference bacteria. For each bacterium, CLSI determines the MIC quality control range for different antibiotic. In this study, the positive controls used were vancomycin for E. faecalis ATCC 29212, where the MIC of 4 μg/mL was obtained (CLSI reference MIC values between 1 and 4 μg/mL) and oxacillin for S. aureus ATCC 29213, where the MIC of 0.2 μg/mL was obtained (CLSI reference MIC values between 0.12 and 5 μg/mL). We do not give a MIC of VAN for E. faecalis B3/101 because it is resistant to vancomycin and there is no MIC reference value by CLSI. The same reason is applied for MRSA S. aureus 74/24 which is resistant to oxacillin. The characterization of multidrug-resistant strains (such as E. faecalis B3/101 and S. aureus 74/24) is described in the previous articles which are found in the reference list. If reviewer #2 needs more detailed explanation, reviewer #2 can read the guidelines from CLSI (Reference 34),

Reviewer #2

Minor:

4, Lines 117-121, Page 4, “The HMBC spectrum (Table 1, Figure S7), exhibited correlations from OH-8 to the carbons at δC 164.5 (C-8), 101.3 (C-7) and 100.0 (C-8a), H-5 to the carbons at δC 166.2 (C-6), C-7, C-8a, C-4, H-7 to C-5, C-6, C-8a, H-3 to Me-9, H-4 to C-4a (δC 142.1), C-5, C-8a, OMe-6 to C-6, Me-9 to C-3, C-4, and a weak correlation from 120 OH-4 to C-3 and C-4.” The description of HMBC correlations of 2 is difficult to read. Please re-write it.

Reply: This description of HMBC correlations is a standard of reporting the correlations from key protons (OH-8, H-5, H-3, H-4OMe-6 and Me-9) to the carbons to support the proposed structure. The description is also accompanied by the data in Table 1. Therefore, I do not see any difficulty in reading this paragraph.

Intriguingly, reviewer #2 made two contradictory comments. In comment 2, reviewer #2 argued that the discussion is redundant because we described the type of protons and carbons and only the data in the table are enough. In this comments, reviewer #2 said that it is difficult to read this paragraph. If reviewer #2 reads the paragraph and look at Table 1, it is much more simple than looking at the table alone.

Reviewer #2

5, The figures for 2D NMR correlations of 2 should be provided in the manuscript.

Reply: The structure of 2 a very simple structure, and the NMR data of its planar structure have been fully discussed in the literature. The novelty of this compound is the absolute configurations of its stereogenic carbons which are different from the previously reported compound and which we uncovered by observing the NOESY correlations and confirmed them by comparison of the experimental and calculated ECD spectra. There are only a few 2D correlations which are well described in Table 1. Thus, we consider adding another figure showing correlations of a simple molecule is not only redundant but also does not add any important information.

Reviewer #2

6, Please clarify whether compound 4 is the first example of 3-alkylpyridinium anthraquinone reported from the nature or just the first example from marine fungus? If it is the latter case, the authors should cite the literatures that reporting other pyridinium anthraquinone-like natural products.

Reply: We wish to thank reviewer #2 for calling our attention. We have changed from “Therefore, 4 is the first 3-alkylpyridinium anthraquinone reported from a marine sponge-associated fungus” to “Therefore, 4 is the first 3-alkylpyridinium anthraquinone reported from the nature.

Reviewer #2

7, Table 4, “1.36, sex (7.4)” ? or should be “1.36, sextet (7.4)” ? Can you really see the sextet coupling for H-14 in the 1H NMR spectrum of 5b? The same issue for H-2’’ of compound 4 in Table 2.

Reply: First, I presume reviewer #2 refers Table#3 and not Table#4. To be in line with the rest, we continue using the abbreviation instead of full name. In this case, we corrected the typo “sex” to a correct abbreviation “sext” for a sextet.

In replying your query “Can you really see the sextet coupling for H-14 in the 1H NMR spectrum of 5b?”, I wish to make it clear that all the data presented in this manuscript are authentic and not invented. I suggest reviewer#2 to check all the spectra in the Supplementary Materials before making any insinuation of the authenticity of the data. Of course, in some cases, the coupling constants are calculated from the expansion of the 1H NMR spectrum of particular zones of interest and were not in the Supplementary Materials.

This principle is also used to present the data for compound 4 in Table 2.

Reviewer #2

8, Figure 1, the putative stereochemistry of 5b and 5c should be shown in the structures as the authors stated that “since 5b and 5c are both dextrorotatory, and based on their biogenic consideration, the configurations of C-16, C-17, C-18, C-19 and C-20 in both compounds should be the same”.

Reply: I think reviewer#2’s comment does not make a minimum sense in asking to show the stereochemistry of the stereogenic carbons when it was clearly stated in the discussion that it was not possible to determine the absolute configurations of the stereogenic carbons by either X-ray analysis or calculated ECD spectrum. The rotations of the compounds are the fact which are obtained by experiment. They cannot be used to indicate the stereochemistry of the compounds.

The second point is that 5c is a C-21 acetate of 5b. Therefore, it is logic to assume that they are derived from the same biosynthetic routes. That does not mean one needs to know or speculate the stereochemistry as the biosynthesis of these compounds have never been carried out and the stereochemistry of the stereogenic carbons have never been elucidated.

Reviewer #2

9, Figure 5 should be polished to make sure the arrows are drawn toward to the correct atoms.

Reply: I have checked all the arrows in Figure 5 and all of them are correctly pointing to the target carbons. Moreover, the HMBC data also are clearly expressed in Table 3 and offer no doubt to which carbons each proton was correlated to. I am really perplexed with this comment.

Reviewer #2

10, The contents of Conclusion are redundant as they are more likely Results, particularly for the conclusion of antimicrobial assays. Please re-write the Conclusion and make it concise.

Reply: I completely disagree with reviewer#2’s opinion that the conclusion is redundant and needs to be rewritten. The conclusion is for the readers to quickly understand what were done in this 20 pages of the manuscript. Therefore, the conclusion is divided into two parts. First, we summarized the compounds isolated from the culture extract of the fungus, highlighting the structural uniqueness of compound 4 as well as emphasizing the structural diversity in the level of stereochemistry of compound 2 besides stating that we have determined the absolute configurations of the stereogenic carbons by X-ray crystallography of the two well-known compounds for the first time.

In the antibacterial assays, we have not only summarized the results we obtained from the isolated compounds but also pointed out the effect of the bulky and polar substituents of the alkenyl resorcinols on the antibacterial and antibiofilm activities of the compounds.

There is no hypothesis, no speculation and no conjecture in the conclusion like I have seen in many papers. Therefore, I don’t think the conclusion is redundant and should be reformulated.

My last note is that after working in this field for 40 years and served as Associate Editor of Marine Drugs since 2014, I have seen many comments from different reviewers of different categories, cultures and geographical zones. In my opinion, reviewers should not give his/her opinion on the subject that he or she doesn’t have expertise (as I have already pointed out above) or insinuate that the authors have made any fraudulent statements (without checking carefully the data) or even making superfluous and unconstructive comments without reading carefully the whole manuscript.

This is exemplified by the huge contrast in opinions of the reviewers for this manuscript.